# Symptom-Based Dispatching in an Emergency Medical Communication Centre: Sensitivity, Specificity, and the Area under the ROC Curve

**DOI:** 10.3390/ijerph17218254

**Published:** 2020-11-09

**Authors:** Robert Larribau, Victor Nathan Chappuis, Philippe Cottet, Simon Regard, Hélène Deham, Florent Guiche, François Pierre Sarasin, Marc Niquille

**Affiliations:** Department of Anaesthesiology, Division of Emergency Medicine, Clinical Pharmacology, Intensive Care and Emergency Medicine, Geneva University Hospital, Rue Gabrielle-Perret-Gentil 4, CH 1211 Geneva 14, Switzerland; Victor.Chappuis@etu.unige.ch (V.N.C.); pinfico@gmail.com (P.C.); simon.regard@hcuge.ch (S.R.); helenetina@wanadoo.fr (H.D.); florent.guiche@hcuge.ch (F.G.); francois.sarasin@hcuge.ch (F.P.S.); marc.niquille@hcuge.ch (M.N.)

**Keywords:** emergency medical dispatch, communication centre, symptom, triage scale, criteria-based dispatch, paramedical, medical priority dispatch, emergency medicine, Fagan nomogram

## Abstract

Background: Measuring the performance of emergency medical dispatch tools used in paramedic-staffed emergency medical communication centres (EMCCs) is rarely performed. The objectives of our study were, therefore, to measure the performance and accuracy of Geneva’s dispatch system based on symptom assessment, in particular, the performance of ambulance dispatching with lights and sirens (L&S) and to measure the effect of adding specific protocols for each symptom. *Methods:* We performed a prospective observational study including all emergency calls received at Geneva’s EMCC (Switzerland) from 1 January 2014 to 1 July 2019. The risk levels selected during the emergency calls were compared to a reference standard, based on the National Advisory Committee for Aeronautics (NACA) scale, dichotomized to severe patient condition (NACA ≥ 4) or stable patient condition (NACA < 4) in the field. The symptom-based dispatch performance was assessed using a receiver operating characteristic (ROC) curve. Contingency tables and a Fagan nomogram were used to measure the performance of the dispatch with or without L&S. Measurements were carried out by symptom, and a group of symptoms with specific protocols was compared to a group without specific protocols. *Results:* We found an acceptable area under the ROC curve of 0.7474, 95%CI (0.7448–0.7503) for the 148,979 assessments included in the study. Where the severity prevalence was 21%, 95%CI (20.8–21.2). The sensitivity of the L&S dispatch was 87.5%, 95%CI (87.1–87.8); and the specificity was 47.3%, 95%CI (47.0–47.6). When symptom-specific assessment protocols were used, the accuracy of the assessments was slightly improved. *Conclusions:* Performance measurement of Geneva’s symptom-based dispatch system using standard diagnostic test performance measurement tools was possible. The performance was found to be comparable to other emergency medical dispatch systems using the same reference standard. However, the implementation of specific assessment protocols for each symptom may improve the accuracy of symptom-based dispatch systems.

## 1. Introduction

Emergency medical dispatching is the first step in emergency medical care [1], the aim being to (1) best allocate available response resources [2] and (2) assist callers in providing on-site first aid to patients [3]. Amongst the many emergency medical dispatch systems, the majority of those in Western and Asian countries are based on a protocol driven approach (algorithm-based), with the most common protocol system being the medical priority dispatch (MPD) system [4,5,6].

In Western Europe, criteria-based systems (CBDs) (guideline driven approach) are more common, especially in EMCCs staffed by nurses or paramedics [7,8]—although the French system is based on a physician-led approach [5]. These emergency medical dispatch concepts [4,9] were developed to assist non-physician emergency medical dispatchers (EMD) in their assessment and prioritization of emergency calls. The MPD system and CBD systems, used by emergency medical communication centres (EMCC) staffed by nurses, paramedics, or non-medical personnel, have moderate sensitivity and, above all, a moderate specificity to detect severe illness or injury [10,11,12].

Emergency medical dispatch systems are not always efficient due to the following factors. First, the two most commonly used systems in Europe (except France) and Western countries (MPD and CBDs), do not completely distinguish between the context of the intervention and the patient’s symptoms. Indeed, assessment protocols are based on a single keyword, which may cover very different symptoms or situations, e.g., keywords, such as “sick person” [11] or “traffic accident” [13], are very imprecise. Secondly, the risk factors associated with the main symptom presented by the patient during the emergency call are poorly known [11,12], not standardized [14], and are poorly integrated into the assessments carried out during emergency calls [15]. In particular, the pre-test probabilities (prevalence) of severe patient conditions encountered in the field are rarely reported accurately. Without accurate measurement and classification of the prevalence (pre-test probabilities) of severe patient conditions, neither the positive or negative predictive values nor the rates of under- or over-triage are interpretable. Finally, measurement is also imprecise because there is no consensus on common standards for reporting, particularly on the reference standard against which to compare the dispatch decision [16].

The decision, made during the emergency call, as to whether or not to send an ambulance with lights and sirens (L&S), can have important consequences both for the ambulance team and for the victim: the use of L&S by the ambulance team greatly increases the risk of traffic accidents [17], whereas significant delays in response may result in patient death [18]. As emergency medical dispatch systems tend to be inefficient, EMCCs frequently apply the rule, “when in doubt, send an ambulance with L&S”, to ensure patient safety, which in turn leads to a high rate of over-triage [11,12,19]. The over-triage rate (100%−positive predictive value in percent) was measured at 71% in CBD systems [12,20], and 80%–94% in MPD [10,11], when the prevalence of a severe patient condition in the field was similar.

In early 2013, our emergency medical communication centre developed a new emergency medical dispatch system, based on two main concepts: (1) a clear distinction between the context of the intervention and the symptom assessment; (2) a definition and symptom assessment adapted from the Swiss emergency triage scale (SETS) used by nurses for patient triage in emergency departments [21,22] and, more recently, by paramedics in the field for referring patients to emergency services [23].

Performance measurement of emergency medical dispatch systems is usually targeted at a specific symptom (e.g., stroke or cardiac arrest) within a specific system. There are very few global measurements, and there is no consensus as to which reference standard to use [16]. Our hypothesis was that it was possible to measure the performance of the Geneva symptom-based dispatching system using the typical tools for measuring the performance of diagnostic tests. The objectives of our study were, therefore, to measure: (1) the overall performance of the symptom-based dispatch system by comparing the levels of the dispatch priorities to the severity of the situations encountered in the field; (2) the performance of the system in terms of dispatching ambulances requiring the use of L&S; and (3) the effect of symptom-specific protocols on triage performance.

## 2. Materials and Methods

This report follows the standards for the reporting of diagnostic accuracy studies (STARD) statement guidelines for reporting diagnostic studies [24].

### 2.1. Settings

The canton of Geneva, covering an area of 282.48 km^2^, is essentially an urban canton, with a population of 493,706 in 2016. Twenty-one per cent of Geneva’s residents were under the age of 20, 16% were over the age of 64 years, and 52% were women. In addition, in 2016, there were approximately 100,000 cross-border workers commuting daily from France or neighbouring cantons to work in Geneva [25].

#### 2.1.1. Geneva’s EMCC

Geneva’s EMCC receives all emergency calls for the canton of Geneva, handling over 68,000 calls per year. Registered nurses and certified paramedics with at least five years field experience staff the EMCC. They are called emergency medical dispatchers (EMD). Geneva’s EMD handles all calls from the beginning (interview) to the end (dispatching). Geneva’s EMDs have evaluated situations using a Symptom-Based Dispatch (SBD) system since early in 2013.

#### 2.1.2. Symptom-Based Dispatch (SBD) System

The principles of the SBD system are described in Figure 1. The first step is to evaluate the event context, i.e., the type of event (malaise or accident), the surroundings (home, public road, etc.), and finally the risks associated with the event or surroundings. The second step is to assess the patient’s state of consciousness, and the third step is to assess their breathing quality. This third step can be bypassed if another symptom assessment protocol is more relevant than the assessment protocols for dyspnoea or coma. The initial assessments regarding the patient’s state of consciousness and breathing quality are intended to quickly detect life-threatening emergencies, particularly cardio-respiratory arrest. The fourth step is to choose the most relevant symptom from a list of 53 symptoms. Finally, Geneva’s EMDs must determine one of the five triage levels on a sorting scale adapted from the SETS.

Although the SETS is composed of four main triage levels, SETS Level 1 (the most severe) is split into two (Level 1-A and Level 1-B). Geneva’s SBD system is still under development and does not yet have a specific assessment protocol for each symptom (Figure 1). Of the 53 symptoms that can be selected at the time of the initial call, there are only 11 symptom-specific protocols currently in use. In most situations, once the main symptom has been selected, the EMD must then determine one of the five triage levels in the sorting scale adapted from the SETS (Figure 2), using only their knowledge and experience of SETS. If there is a specific protocol for symptom assessment (the 11 specific protocols), the emergency medical dispatcher must choose a single determinant from those listed to define the level of triage, beginning the assessment with the most serious to least serious determinant.

#### 2.1.3. Geneva’s EMS

Geneva’s EMS is a two-tier system. First level response ambulances assigned to priorities 1 and 2 are staffed by two paramedics, while ambulances assigned to priority 3 are staffed by a paramedic and an emergency medical technician. Paramedics have approximately thirty official protocols related to the autonomous treatment of all symptoms. They are trained to set up intravenous or intraosseous access, administer emergency medication, and perform all advanced cardiopulmonary resuscitation measures but not oro-tracheal intubation. The second level of response involves the pre-hospital emergency physician who can be sent on site (by ground or helicopter) simultaneously with the ambulance, or later, at the paramedics’ request. The presence of a senior physician, who may be dispatched to the second line following a request from the on-site team, creates a subsidiary third tier. The paramedic or pre-hospital emergency physician then assesses the patient’s condition on site in accordance with the National Advisory Committee for Aeronautics (NACA) scale [26] and informs the EMCC before leaving the intervention site.

### 2.2. Reference Standard

The NACA scale is used in Switzerland [12] and other European countries [27] in pre-hospital medicine to assess a patient’s condition in the field and is significantly correlated with survival [26,28]. The NACA scale is calculated when the patient’s most severe condition is observed during the intervention in the field. The NACA score is a seven-level symptomatic scale. The seven levels of the NACA score are defined in Figure 2. We divided level seven into two to distinguish between a deceased patient, for whom there was no attempt at resuscitation (NACA 7 no-res), and a patient on whom resuscitation was attempted (NACA 7 res).

The NACA scores were dichotomized into two groups, one where the severity of the symptoms observed is critically time-dependent (time-critical patients, NACA ≥ 4), and the other where the symptoms are less critical (less-time-critical patients, NACA < 4). We chose this reference standard because there is no universally accepted reference standard and the NACA scale is the reference scale used in the field in Switzerland. We chose this dichotomization threshold of the NACA scale because other authors have used this in a similar emergency dispatching systems [12,20] and because a patient who presents with a condition that may eventually lead to the deterioration of vital signs is most likely a time-critical patient.

Figure 2 shows the correspondence between the five levels of triage defined by the symptom assessment and the three levels of ambulance dispatch priorities (SBD priority system). In Switzerland, L&S are only used for priority 1. When the triage level is 3 or 4 (priority 3), Geneva’s EMD may decide not to send an ambulance to the site, but to propose an alternative response, such as contact with a general practitioner.

### 2.3. Selection of Participants

Data from all emergency calls are collected in Geneva’s EMCC computer-aided dispatch (CAD) software system. The SBD system was set up at the beginning of 2013 and the CAD software system was considered stabilized as of 1 January 2014. Since that date and until 30 June 2019, all evaluations (with the exception of inter-hospital transfers), for which a main symptom and a dispatch priority were defined, were included in our study. When a patient is identified during the initial call, the CAD software system obliges EMCC staff to define a symptom and dispatch priority before an ambulance can be dispatched. As emergency medical dispatching systems differ from region to region, we decided to include only the Geneva EMCC assessments, thus, using a convenience sample rather than a sample size calculation.

### 2.4. Study Design and Measurements

This was a prospective observational diagnostic study that included data from 1 January 2014 to 1 July 2019.

First, the receiver operating characteristic (ROC) curve of the dispatch levels to predict NACA ≥ 4 were reported to measure the overall performance of the SBD system. Second, the prevalence of symptoms defined during the emergency call and the severity prevalence of situations observed in the field were measured. Third, dispatch priorities were reported for each symptom defined during the initial phone call. Fourth, the NACA score observed by EMS staff in the field was reported for each symptom.

Dispatch priority was dichotomized into: (1) priority with the use of L&S (Priority 1); and (2) priority without the use of L&S (Priority 2 and Priority 3). The “diagnostic test” as to the choice between the dichotomized dispatch priorities (use of L&S/no use of L&S) was compared to the symptom severity reference standard (time-critical patients, NACA ≥ 4/less-time-critical patients, NACA < 4). The sensitivity and specificity, Positive Predictive Value (PPV) and Negative Predictive Value (NPV), over-triage rate (100%−predictive positive value in percent) and under-triage rate (100%–predictive negative value in percent), were measured for the choice of dichotomized dispatch priority of each symptom defined during the initial call. The main results are highlighted in a Fagan nomogram.

Geneva’s SBD system does not yet have a specific assessment protocol for each symptom (Figure 1). Of the 53 symptoms that may be selected during the initial call, there are only eleven specific symptom protocols currently in operation. Therefore, we also compared the dispatch priority choices (using L&S/without L&S) made using specific protocols to those made without specific protocols. We compared the group of symptoms assessed using specific protocols to the group of symptoms assessed without specific protocols. We assessed the influence of “spectrum bias” [29] on the symptom groups by calculating the likelihood ratio of the test result in the groups divided by the likelihood ratio for the same test result in the overall patient population.

### 2.5. Statistical Analysis

The CSV file containing the data from the Geneva SBD system was imported into the STATA^®^ 14.2 software (StataCorp^®^, College Station, TX, USA). ROC curves, prevalence, sensitivity, specificity, predictive values, and their 95% confidence interval calculations, as well as all descriptive statistic calculations, were performed using STATA^®^ 14.2 software. The Chi2 test (STATA 14.2) was used to compare the proportions of categorical variables. A test was considered significant when *p* < 0.05.

### 2.6. Ethics

Approval of the study was given by the Cantonal Commission for Research and Ethics of Geneva (project n 2018-00789) on 12 June 2018.

## 3. Results

We considered 194,764 primary evaluations for this study. We excluded 10,870 evaluations because there was no documentation setting out the main symptom or dispatch priority. In addition, we excluded 34,915 (19.0%) interventions for which the NACA scale was not reported (see Appendix A—Supplemental (1)). There were, therefore, 148,979 documented assessments of symptoms during the 5.5 years of the study (flow chart of the study participants is given in Appendix A (Supplemental (2)).

The incidence of level 1 dispatch priorities (with L&S) was 32.7, with a 95% CI (32.2–33.1) per 1000 inhabitants per year (calculation for an average annual population of 550,000, including interventions for which the NACA scale was not reported).

The area under the receiver-operating characteristics curve (AUROC) of 0.7474, 95% CI (0.7448–0.7503) shows an acceptable triage ability for the SBD system (Figure 3 below). However, sensitivities and specificities varied in the opposite direction depending on the level of triage. The more severe the level of triage, the higher the specificity, while the sensitivity decreased.

Overall, when the EMD assesses a situation during an emergency call, the pre-test probability that the severity of the situation found in the field corresponds to a NACA ≥ 4 is 21.0% with a 95% CI (20.8–21.2). If the EMD decides to dispatch an ambulance with L&S, the post-test probability (of NACA ≥ 4) is 30.6%, 95% CI (30.3–30.9). If the EMD decides not to dispatch an ambulance with L&S, there will be 6.6%, 95% CI (6.4–6.8) in the field situations corresponding to a NACA ≥ 4 (Figure 4 below). Thus, it is mainly the decision to not dispatch an ambulance with L&S that will best predict the finding of a small proportion of severe patients in the field (negative likelihood ratio < 0.5).

In Table 1, the rate of severe situations encountered in the field was much lower than the rate of ambulances dispatched with L&S. As a result, the rate of over-triage was high (69.4%, 95% CI (69.3–69.9)). Consequently, we observed that, logically, the positive predictive values of the different symptom categories varied in the same direction as the severity prevalence values, while the negative predictive values varied in the opposite direction. With the exception of trauma or accident situations, we also noted that the sensitivity rates of ambulance dispatch with L&S depending on the different symptom categories varied in the same direction as the severity prevalence values while the specificity rates varied in the opposite direction. For trauma and accident situations, the rate of over-triage was extremely high (89.1% CI 95% (89.5–90.3)) while the severity prevalence remained low. Details for all symptoms are given in Appendix A (Supplemental (3)). The sensitivity was greater than 80% for 22 of the 53 symptoms assessed. The specificity was greater than 80% for only 9 of the 53 symptoms assessed.

When we compared the assessments performed using protocols, we found that the level of sensitivity was better (*p* < 0.001), whilst the specificity was better (*p* < 0.001) when the assessment was not performed using protocols (Table 2). However, we noted that the severity prevalence observed in the field (NACA scale > 4) was twice as high in the group where protocols were used. Finally, we found no significant “spectrum bias” for the symptom groups evaluated using specific protocols compared to the symptoms evaluated without specific protocols.

## 4. Discussion

Using the dichotomized NACA scale (NACA cut-off ≥ or < 4) as a reference standard when measured in the field, we were able to construct a ROC curve with the five triage levels defined during the emergency call. The global AUROC of 0.7474, 95% CI (0.7448–7503) showed an acceptable triage capacity for the SBD system. To our knowledge, this is the first overall performance measurement of an emergency medical dispatch system reported in this way. ROC analysis has been extensively used in the evaluation of diagnostic tests [30]. The AUROC measure allows for comparison of the performance of emergency dispatch systems, and this measure is essential for improving the quality of emergency medical dispatch [16]. In addition, in the SBD system, it is possible to construct a ROC curve for each symptom identified during the emergency call and to estimate the contribution to the ROC curve of each of the symptom-defined triage levels. In this way, it is possible to improve the accuracy of the discriminating questions for each symptom and, thus, improve the quality of the triage.

There are studies that compared the indication of ambulance dispatch with L&S to the severity found in the field [11,12,20]. Using the same reference standard as Dami & al, the sensitivity (87.5% (95% CI: 87.1–87.8)) and specificity (47.3% (95% CI: 47.0–47.6)) values we measured were similar to the sensitivity (86.0% (95% CI: 85.6–86.4)) and specificity (48.0% (95% CI: 47.4–48.6)) values measured in their criteria-based dispatching system [12]. In the medical priority dispatch system, the values of sensitivity (93.3% (95% CI: 92.7–93.9)) and specificity (48.7% (95% CI: 48.5–48.9)) were also very close to ours, even if the reference standard used was different. In this respect, we found that the overall performance of our SBD system was similar (not superior) to the CBDs and MPD systems.

Comparing the rates of over-triage and under-triage (and thus the positive and negative predictive values), between dispatch systems, requires measuring the severity prevalence found in the field. Fagan’s nomogram makes it easy to visualize the differences between systems, provided that it is the PPV and the under-triage rate that are represented on the post-test probability axis. The two values on the post-test probability axis then represent the rate of patients with a severe condition found in the field, depending on the choice made during the emergency call (dispatch with or without L&S). The severity prevalences observed in the field reported by Ball [11] (3.3%) and Dami [12] (14%) were different to ours (21%). The post-test probabilities (and thus the values of over or under-triage) are therefore not comparable.

There was a large difference in the severity prevalence between the symptom groups, and logically, the differences found in the positive predictive values of these groups reflect this. In an emergency medical dispatch system, the choice of the main symptom during the emergency call is in itself essential to estimate the post-test probability of severity that will be observed in the field. The higher the severity prevalence of the symptom, the higher the sensitivity and lower the specificity. This is a typical “spectrum effect” (not “spectrum bias”) [29,31,32].

The most severe symptoms, such as respiratory distress, coma, or cardiac arrest, are also likely the “most obvious” and therefore the easiest to recognize during the emergency call [33]. These symptoms are over-represented in the “heart, circulation, and breathing” group, which has the highest severity prevalence. Concerning trauma or accident situations, we believe that the particularly high rate of over-triage is linked to: (1) the non-existence of specific evaluation protocols for each of the trauma symptoms in our SBD system and (2) the confusion between the real severity of the symptoms and the circumstances surrounding the accident, which often requires ambulance dispatch with L&S.

When we compare evaluations performed with protocols with those performed without protocols, we also found a large “spectrum effect” [29] that impacted the sensitivity and specificity in opposite ways. Therefore, we cannot conclude that it was the protocols that affected the sensitivity and specificity. We can only conclude that situations with a high severity prevalence are likely easier to recognize during the emergency call than situations with a low severity prevalence. Differences in positive predictive values between the groups of assessments conducted using protocols and those conducted without protocols are, therefore, only the consequence of differences in the severity prevalence between the two groups. However, regarding the negative predictive values, the absence of differences between the two groups indicates that the implementation of protocols for severe situations did not significantly worsen the under-triage. Conversely, the implementation of specific protocols in our SBD system appeared to improve the accuracy of emergency medical dispatching; this confirms the results of a survey on German EMCCs, concerning the impact of taking structured emergency calls [34].

There are several limitations to our study. It was a single-centre study, which may limit the generalisability of our results. There were a significant number of assessments during the emergency call for which the NACA scale was not documented. These were mostly non-severe situations, often with the main symptoms “other”, “unspecified malaise”, or “Person lying, without possibility to evaluate”, where either an ambulance was not dispatched or an ambulance was dispatched but did not transport the patient. Thus, a bias in favour of severe situations is possible. The choice of the reference standard used is questionable, notably because the dichotomization threshold of the NACA scale may not be the right one. Indeed, in a similar EMCC, the severity prevalence observed in the field according to the NACA scale was significantly different [12].

Choosing a reference standard for “life-threatening emergency in the field” versus “non-life-threatening emergency”, as used by Ball [11], may have allowed for a better comparison between EMCCs. However, this was a large sample study that highlights key elements for measuring the performance of emergency medical dispatching. To compare the assessment qualities between EMCCs, it is essential to distinguish the context of the symptoms presented by patients during emergency calls. In addition, the severity prevalence of each symptom must be measured to interpret and clinically use the “diagnostic tests”. The Fagan nomogram is, thus, particularly useful in the Bayesian analysis that is performed during the telephone medical assessment. Finally, the presentation of the evaluation results during the emergency call in the form of a ROC curve makes it possible to immediately visualize the contribution of each level of dispatch in the performance of the evaluation.

## 5. Conclusions

Geneva’s SBD system was developed specifically to assist nurses and paramedics in the assessment of emergency calls. It is a mixed system between the protocol-driven system and the guideline-driven system.

The performance of this dispatch system specifically based on symptom assessment (the SBD system) does not appear to be superior to other emergency medical dispatch systems that use the same reference standard. Accurately identifying the symptoms presented by patients during the emergency call and measuring the severity prevalence for each symptom are essential for interpreting predictive values and comparing emergency medical dispatch systems. Implementing specific assessment protocols for each symptom can improve the accuracy of the symptom-based dispatch system. Therefore, we propose that the development of new specific assessment protocols for each of the 53 symptoms should be continued and assessed.

The performance of emergency medical dispatching should be comparable to a universally agreed reference standard. The “dichotomized NACA scale” reference standard remains imperfect. It should be possible to compare the assessment scale used during the emergency call with an identical scale once the ambulance arrives at the intervention site.

Describing the symptoms presented by patients during emergency calls and measuring the quality of the assessments made should be the rule in emergency medical communication centres, not the exception. Future research is required to specify the reference standard and describe the rules to be applied to compare different emergency medical dispatching systems in a reliable manner.

## Figures and Tables

**Figure 1 ijerph-17-08254-f001:**
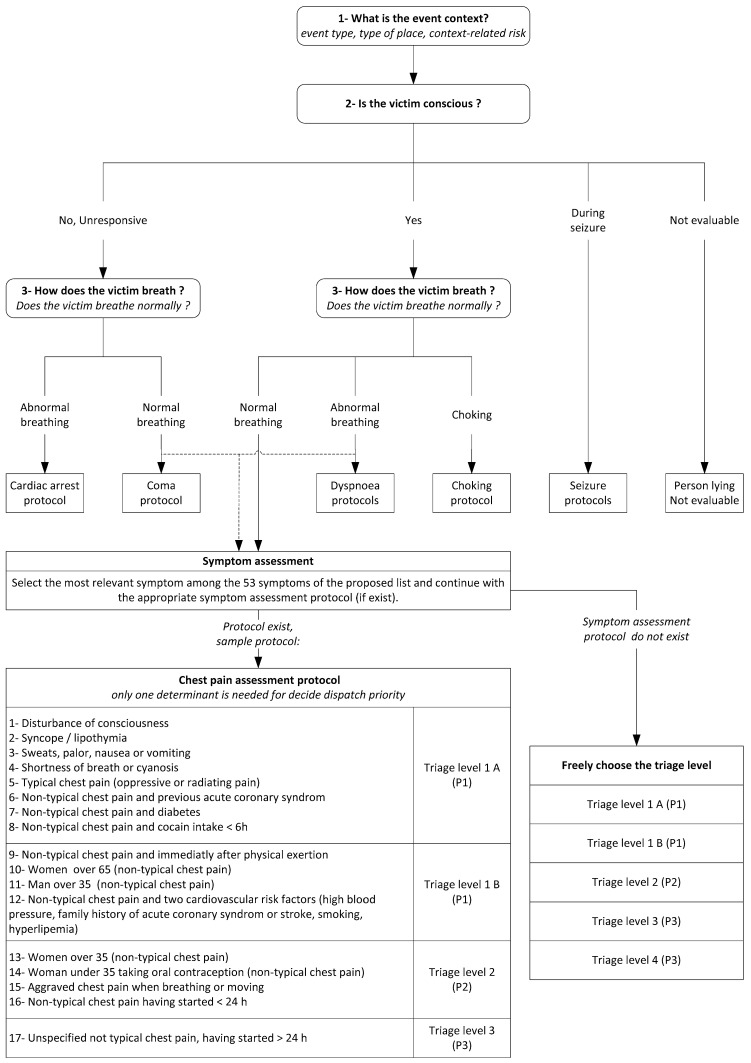
Principles of the symptom-based dispatch system.

**Figure 2 ijerph-17-08254-f002:**
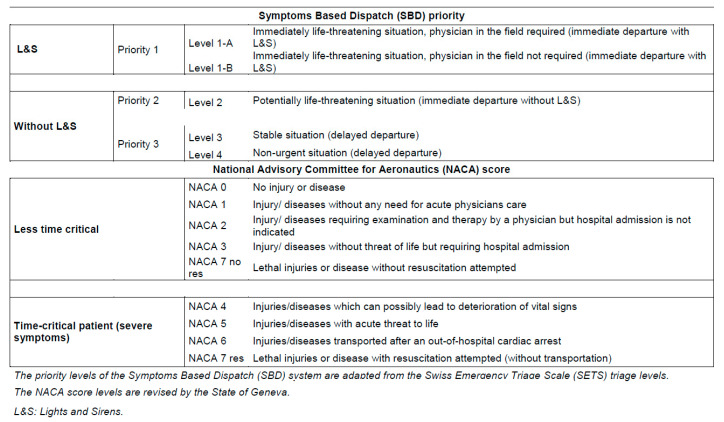
Definition of the dispatch priority levels and the NACA score levels on the field.

**Figure 3 ijerph-17-08254-f003:**
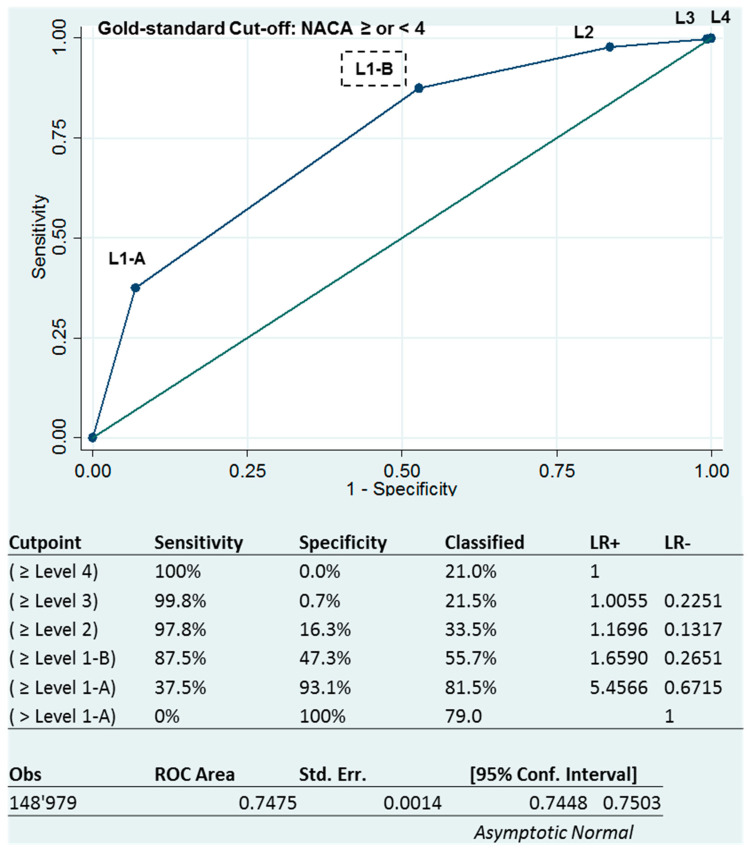
Receiver operating characteristic (ROC) curve of the dispatch levels to predict NACA ≥ 4.

**Figure 4 ijerph-17-08254-f004:**
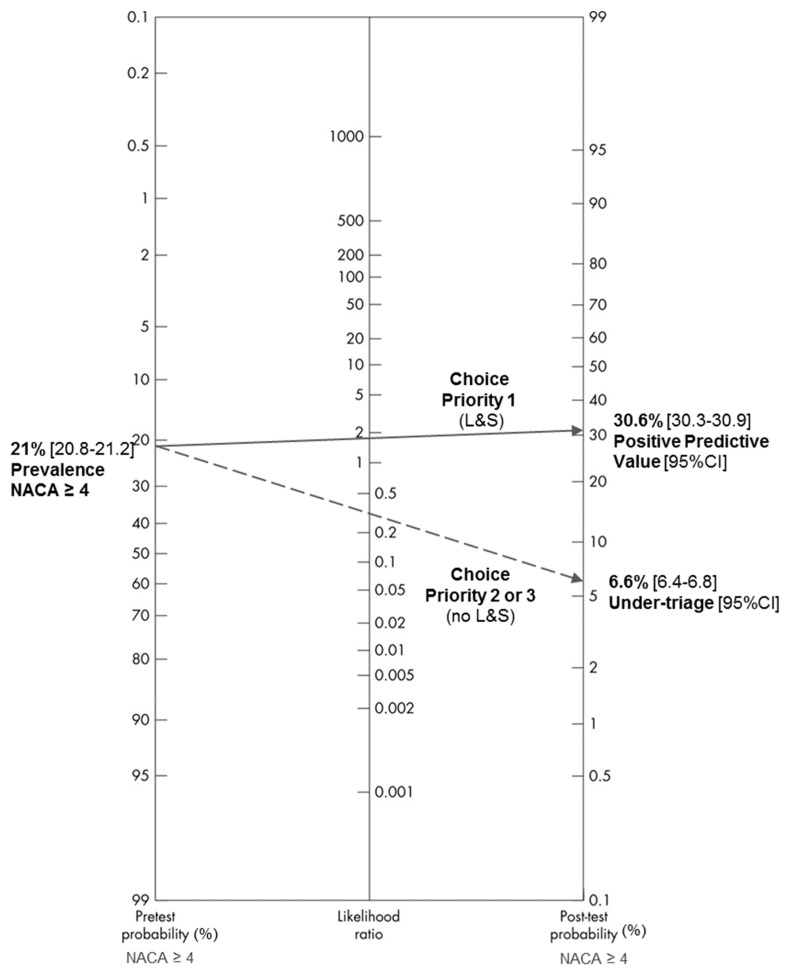
Fagan nomogram of the dispatch with or without L&S in relation to the dichotomized NACA scale (NACA ≥ or < 4).

**Table 1 ijerph-17-08254-t001:** Priorities for dispatching ambulances with L&S compared to NACA situations ≥ 4 found in the field, by symptom categories.

	All EVALUATIONS n (%)	NACA Scale ≥4 (Severe Symptoms)	Dispatch Priority P1 (with L&S)	Sensitivity (95%CI)	Specificity (95%CI)	PPV (95%CI)	NPV (95%CI)
**All Symptoms evaluations**	148,979	31,269 (21.0)	89,410 (60.0)	87.5 [87.1–87.8]	47.3 [47.0–47.6]	30.6 [30.3–30.9]	93.4 [93.2–93.6]
**Age, years** (±SD)	58.9 (±26.3)	64.5 (±24.0)	56.2 (±26.6)				
**Male sex**	70,726 (47.5)	15,896 (22.5)	44,044 (62.3)				
**Female sex**	71,839 (48.2)	14,206 (19.8)	40,777 (56.8)				
**Heart, circulatory and respiratory**	30,906 (20.7)	13,790 (44.6)	26,518 (85.8)	95.0 [94.6–95.3]	21.6 [21.0–22.2]	49.4 [48.9–50.0]	84.3 [83.1–85.3]
**Neurological, psychiatrics**	37,658 (25.3)	9036 (24.0)	22,919 (60.9)	86.1 [85.3–86.8]	47.1 [46.5–47.7]	33.9 [33.3–34.6]	91.5 [91.0–91.9]
**General symptoms, other**	32,452 (21.8)	4934 (15.2)	16,470 (50.8)	79.1 [78.0–80.3]	54.3 [53.7–54.9]	23.7 [23.1–24.2]	93.6 [93.2–93.9]
**Abdominal, Urological, gynaeco-obstetrical, back**	13,191 (8.9)	961 (7.3)	3282 (24.9)	55.0 [51.8–58.2]	77.5 [76.7–78.2]	16.1 [14.9–17.4]	95.6 [95.2–96.0]
**Trauma, Accidents**	34,772 (23.3)	2548 (7.3)	20,221 (58.2)	80.0 [78.4–81.6]	43.6 [43.0–44.1]	10.1 [9.7–10.5]	96.5 [96.2–96.8]

All variables given as numbers (group percentages in parenthesis) except age. Sensitivity, specificity, PPV, NPV, are shown in percentage; L&S: Light and Siren; CI: Confidence Interval; PPV: Positive Predictive Value; and NPV: Negative Predictive Value.

**Table 2 ijerph-17-08254-t002:** Impacts of the protocols on the accuracy on the assessment.

	All Evaluations	NACA Scale ≥ 4 (Severe Symptoms)	Dispatch Priority P1 (with L&S)	Sensitivity (95% CI)	Specificity (95% CI)	PPV (95%CI)	NPV (95%CI)
**No protocol**, n (%)	119,899	20,364 (17.0)	67,652 (56.4)	83.6 [83.1–84.2]	49.1 [48.8–49.5]	25.2 [24.9–25.5]	93.6 [93.4–93.8]
**Protocol**, n (%)	29,080	10,905 (37.5)	21,758 (74.8)	94.6 [94.2–95.0]	37.2 [36.3–37.8]	47.4 [47.1–48.5]	92.0 [91.3–92.6]
***p*** **value**				*< 0.001*	*< 0.001*	*< 0.001*	*0.602*

CI: Confidence Interval; PPV: Positive Predictive Value; NPV: Negative Predictive Value. Bold shows that the threshold of the *p*-value is much lower than 0.05.

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
