# Peer review of "Symptom-Based Dispatching in an Emergency Medical Communication Centre: Sensitivity, Specificity, and the Area under the ROC Curve"

_ijerph, 2020, doi:10.3390/ijerph17218254_

Round 1

Reviewer 1 Report

The large number of calls excluded in the analysis because the NACA scale was not done is concerning and may have biased the results.  Since the NACA scale is the gold standard, shouldn't it be calculated for every call regardless of the disposition of the call?  This warrants further discussion.  Is is possible to obtain any data on the calls where the NACA scale was not calculated so that these calls can be compared with the analytic sample?

Author Response

1- The large number of calls excluded in the analysis because the NACA scale was not done is concerning and may have biased the results. 

Response 1: This is true, but it is unlikely to have had a significant influence on the main outcome (performance of sending the ambulance with lights and sirens), as most of the situations where the NACA scale was not reported concern situations where an ambulance was not dispatched or an ambulance was dispatched but did not transport the patient, that is, situations of low severity (P2-P3 =73% of total NACAs not reported, see table supplemental 1). Furthermore, these are mostly (non-severe) situations, often with the main symptoms "other", "unspecified malaise", or "Person lying, without possibility to evaluate”.

2- Since the NACA scale is the gold standard, shouldn't it be calculated for every call regardless of the disposition of the call?  This warrants further discussion. 

Response 2: Indeed, we have completed the discussion on this topic. Lines 342-346

3- Is is possible to obtain any data on the calls where the NACA scale was not calculated so that these calls can be compared with the analytic sample?

Response 3: We have added a new table "supplemental 1".

Reviewer 2 Report

The paper addresses a really interesting and important issue and overall, the authors already did a good job with the paper. It is well-written and good to follow. I have a few remarks I would ask the authors to address when revising their work.

Abstract:

  • Was exactly is the gold standard and to which other systems is the dispatching system not superior to? (Same in line 56.) Later, the paper calls the NACA score the gold standard, but this is not clear at the beginning. Overall, I am not a fan of the word. While it is highly accepted and used and studied, is it really the holy grail? The authors rightly discuss it themselves in Sec. 4. It might be good to already comment it briefly in Sec 2/3
  • Before evaluating superiority, would it not be the more important to independently analyse the dispatch performance?

Introduction:

  • Lines 41/42: most used dispatching systems, worldwide? Despite differences in EMS systems worldwide?
  • Shouldn’t there be more relevant literature on the topic?

Section 2:

  • Lines 91-103: this paragraph could be a bit clearer and a bit more detailed.
  • Figure 1: The symptom assessment does not really become clear; How does the following work: “Select the most relevant symptom among the 53 symptoms of the proposed list and continue with the appropriate symptom assessment protocol (if exist).”
  • Figure 1: how were the assessments defined? Are there references for it? E.g., why the age limits – woman over 65 or man over 35?
  • Sec 2.1.2: the authors should bring this earlier
  • Sec 2.3: this would also be good to have earlier in the paper

Section 3:

  • The authors say in line 139 that “The NACA scale is calculated when the patient’s most severe condition is observed during the 139 intervention in the field.” If this is the case at the end of treatment at the scene, shortly before the patient is transported, can this score really be used to validate the L&S decision?
  • Line 230: Negative Predictive Value

Section 5:

  • This is too short. What are the conclusions regarding the current dispatch system in general? What could / should be changed? As the authors themselves question the chosen “gold standard”, I think the general observations and potential improvements should be more important than the comparison?!
  • I am missing an outlook on future research at the end of the paper.

References are numbered twice.

Author Response

I- Abstract:

1- Was exactly is the gold standard and to which other systems is the dispatching system not superior to? (Same in line 56.)

Response 1: The reference standard is the dichotomised NACA. This has been specified in the abstract and detailed in section 2 (materials and method, 2.2). Systems to which SBD is not superior are emergency medical dispatch systems that use the same reference standard. Lines 138-151.

2- Later, the paper calls the NACA score the gold standard, but this is not clear at the beginning. Overall, I am not a fan of the word. While it is highly accepted and used and studied, is it really the holy grail?

Response 2: Indeed, we have replaced the word "gold" with "reference" throughout the manuscript. The dichotomized NACA scale is clearly not the Holy Grail. It is the only reference we have, but its relevance is debatable.

3- The authors rightly discuss it themselves in Sec. 4. It might be good to already comment it briefly in Sec 2/3

Response 3: We have clarified this in Section 2, paragraph 2.2. Lines 138-151

4- Before evaluating superiority, would it not be the more important to independently analyse the dispatch performance?

Response 4: That's right, and that's actually what we tried to do. We have modified the hypothesis accordingly in the introduction & abstract. The comparison with other emergency dispatching systems was described in the discussion. Lines 79-83 and lines 29-32.

II- Introduction:

5- Lines 41/42: most used dispatching systems, worldwide?

Response 5: Indeed, we have amended the entire first paragraph of Section 1 (introduction).  We have clearly differentiated between "protocols driven" and "guidelines driven" dispatch systems and explained the main differences between countries, citing four new references (references number 4,6,7,8). Lines 43-56.

6- Despite differences in EMS systems worldwide?

Response 6: We have clearly differentiated between "protocols driven" and "guidelines driven" dispatch systems and explained the main differences between countries in the introduction ), Lines 43-56 

7- Shouldn’t there be more relevant literature on the topic?

Response 7: We have cited four new references (numbers 4,6,7,8). Lines 43-56.

III- Section 2:

8- Lines 91-103: this paragraph could be a bit clearer and a bit more detailed.

Response 8: We have rewritten the relevant paragraph 2.1.2 to make it clearer. Lines 118-126

9- Figure 1: The symptom assessment does not really become clear; How does the following work: “Select the most relevant symptom among the 53 symptoms of the proposed list and continue with the appropriate symptom assessment protocol (if exist).”

Response 9: We have reviewed and rewritten the explanation of Figure 1 in the paragraph 2.1.2. Lines 118-126

10- Figure 1: how were the assessments defined? Are there references for it? E.g., why the age limits – woman over 65 or man over 35?

Response 10: The specific protocol proposed as an example is the one that emergency medical dispatchers use to assess chest pain. The definition of the symptom "chest pain" is that of the Swiss emergency triage scale. The determinants are defined in the Swiss emergency triage scale or have been defined by the medical director of the Geneva emergency medical communication centre. In the example, the cardiovascular risk of a man over 35 years of age is considered equivalent to that of a woman over 65 years of age.

11- Sec 2.1.2: the authors should bring this earlier

Response 11: We have moved paragraph 2.1.2. Geneva's EMS before Figure 2 and before paragraph 2.2 standard reference.

12- Sec 2.3: this would also be good to have earlier in the paper

Response 12: We have moved paragraph 2.3 standard reference before figure 2 and before paragraph 2.2 selection of participants.

Section 3:

13- The authors say in line 139 that “The NACA scale is calculated when the patient’s most severe condition is observed during the intervention in the field.” If this is the case at the end of treatment at the scene, shortly before the patient is transported, can this score really be used to validate the L&S decision?

Response 13: Indeed, this is an excellent question. A patient's clinical condition can vary at any time, with or without care. The clinical condition may be good during the emergency call, then deteriorate before the ambulance arrives in the field. Afterwards, the condition may improve or deteriorate during care, and this may be associated with the care provided or treatment administered. The reference standard should be measured when the care team arrives at the patient's site. In practice, this measurement is often not explicit and is not systematically documented. The NACA scale that is transmitted is the measure of "the patient's most serious condition observed in the field". It is therefore not the state at the end of treatment in the field, even though this scale is usually transmitted by radio at that time. It is therefore the best reference standard we currently have, even if it is not perfect. We assume that most patients will improve as a result of the pre-hospital care provided, and that therefore "the patient's most serious condition observed in the field" is usually the condition observed at the beginning of the field treatment.

14- Line 230: Negative Predictive Value

Response 14: corrected in the text (typographical error)

Section 5:

15- This is too short. What are the conclusions regarding the current dispatch system in general?

Response 15: The conclusion has been completed. It is a mixed system between the protocol-driven system and the guideline-driven system. Lines 361-363

16- What could / should be changed?

Response 16: The conclusion has been completed. The development of new specific evaluation protocols should therefore be continued. The evaluation scale used during the emergency call should be comparable to an identical scale when the ambulance arrives in the field. Lines 369 375

17- As the authors themselves question the chosen “gold standard”, I think the general observations and potential improvements should be more important than the comparison?!

Response 17: The conclusion has been completed. The performance of emergency medical dispatching should be comparable to a universally agreed reference standard. The "dichotomised NACA scale" reference standard remains imperfect. Lines 372-375

18- I am missing an outlook on future research at the end of the paper.

Response 18: The conclusion has been completed. Lines 376-379

19- References are numbered twice.

Response 19: I couldn't find which reference was numbered twice.

Reviewer 3 Report

  1. I would suggest to write in the introduction in which country, city the system would be analysed as it is not mentioned clearly in the introduction section and also - why? 
  2. Some prior research should be mentioned in the introduction section - sth like state of research.
  3. in the method part - the range of research should be underlined in clear way. 

Author Response

1- I would suggest to write in the introduction in which country, city the system would be analysed as it is not mentioned clearly in the introduction section and also - why?

Response 1: We have added "Geneva's" in the abstract to line 14. We have added "Switzerland" in the abstract on line 18. We have amended the entire first paragraph (lines 43-56) of Section 1 (introduction). We have clearly differentiated between "protocols driven" and "guidelines driven" dispatch systems and explained the main differences between countries, citing four new references (references number 4,6,7,8) in the section 1 (introduction). Lines 43-56. It was an oversight.

2- Some prior research should be mentioned in the introduction section - sth like state of research.

Response 2: We have cited four new references (numbers 4,6,7,8) and we have amended the last paragraph of the section 1 (introduction) to clarify these aspects. Lines 82-88.

3- In the method part - the range of research should be underlined in clear way

Response 3: We have clarified this point by restructuring Section 2, in particular the order of the paragraphs, which makes this section more understandable.

The scope of the research and the measurements carried out are now specified in paragraph 2.4. Study design and measurements. Lines 199-223.

Round 2

Reviewer 1 Report

Thank you for submitting a revised version. 

Reviewer 3 Report

Corrections improved the quality of paper